# PrEP discontinuation, cycling, and risk: Understanding the dynamic nature of PrEP use among female sex workers in South Africa

Lillian M. Shipp[1]*, Sofia Ryan[2], Carly A. Comins[1], Mfezi Mcingana[3], Ntambue Mulumba[4], Vijayanand Guddera[5], Deliwe Rene Phetlhu[6], Harry Hausler[3,7], Stefan D. Baral[1], Sheree R. Schwartz[1]

1 Department of Epidemiology, Johns Hopkins Bloomberg School of Public Health, Baltimore, Maryland, United States of America, 2 Health Sciences Department, NORC at the University of Chicago, Bethesda, Maryland, United States of America, 3 TB HIV Care, Cape Town, South Africa, 4 TB HIV Care, uMgungundlovu, South Africa, 5 TB HIV Care, eThekwini, South Africa, 6 Sefako Makgatho Health Sciences University, Ga-Rankuwa, South Africa, 7 Department of Family Medicine, University of Pretoria, Pretoria, South Africa

* lshipp@umich.edu

**Data Availability Statement:** The full dataset cannot be shared publicly due to concerns around participant privacy given the potential for

## Abstract

PrEP cycling among women is thought to be safe when there are distinct "seasons of risk." However, cyclical PrEP use over short periods may be associated with increased risk of HIV acquisition. We aimed to characterize key social ecological factors contributing toward PrEP cycling among female sex workers (FSW) in the context of high HIV risk. Semi-structured, in-depth interviews were conducted with 36 FSW at risk for HIV acquisition and 12 key informant (KI) service providers in eThekwini (Durban), South Africa from January-October 2020. FSW identified key factors driving temporary discontinuation of PrEP including relocation, lack of information on or difficulty coping with side effects, and delays in accessing PrEP. In many cases, FSW were motivated to restart PrEP once barriers were overcome. In contrast, KIs emphasized the importance of individual adherence to PrEP and reliance on personal risk assessments when counselling FSW on cycling decisions. FSW and KI perspectives highlight a disconnect between providers' recommendations on the potential for cyclical use of PrEP during periods of minimal risk and actual drivers among FSW causing temporary PrEP discontinuation. Further interventions supporting safe PrEP cycling are needed to ensure decisions around cycling are deliberate and guided by changes in HIV risk rather than external factors.

## Introduction

South Africa has one of the highest burdens of HIV in the world, with an estimated 8.2 million people living with HIV as of 2021 [1]. Among female sex workers (FSW) in South Africa, the burden is disproportionately high compared to the population overall, with nearly 60% of FSW estimated to be living with HIV [2]. Despite relatively high rates of reported condom use among FSW in South Africa, violence and coercion are prevalent among sex workers and

identification of participants from full interview transcripts. All relevant quotes and demographic information are included in the manuscript. Requests for deidentified data may be sent to Stefan Baral at sbaral@jhu.edu.

**Funding:** This study was funded by the National Institute of Nursing Research [R01NR016650, R01NR016650-04S1] in the form of grants to SB and by the Johns Hopkins University Center for AIDS Research through the National Institutes of Health [P30AI094189]. This study was also funded by the National Institute of Mental Health [R01MH121161] in the form of a grant to SS. The content is solely the responsibility of the authors and does not necessarily reflect the official views of the NIH. The funders had no role in study design, data collection and analysis, decision to publish, or preparation of the manuscript.

**Competing interests:** The authors have declared that no competing interests exist.

often challenge effective condom use negotiation with clients and non-paying partners [3–5]. A recent model of HIV transmission in South Africa indicated that transactional sex between FSW and their clients comprised 7% of the annual new HIV infections in the country from 2010–2019 [6]. Consequently, FSW were prioritized for specific interventions in South Africa's National Strategic Plan for HIV, Tuberculosis (TB), and Sexually Transmitted Infections (STIs) for 2017–2022 [2, 7].

Pre-exposure prophylaxis (PrEP) is an HIV prevention consistently shown to be more than 90% effective in preventing HIV acquisition when taken as prescribed [8]. As an individually controlled HIV prevention method, it shows particular promise as a means of HIV prevention for key populations including cisgender FSW when combined with condom use. Among cisgender women, efficacy trials and demonstration projects have found that oral PrEP is most effective in protecting against HIV when taken as a daily regimen rather than an event or time-driven approach [9–11]. While a long-term, daily regimen is recommended for women during periods of elevated risk, guidelines set out by the Southern African HIV Clinicians Society indicate that cycling on and off of PrEP may be appropriate when driven by changes in HIV acquisition risk [12]. However, in South African PrEP demonstration projects among adolescents [13] and FSW [10, 14], rates of PrEP persistence past one, three, and six months drop significantly despite sustained risk and high HIV incidence [15, 16]. While barriers and facilitators to PrEP uptake and adherence among young women and to a lesser extent FSW have been described [17–20], less is known about whether women who discontinue PrEP choose to later re-initiate and what potential motivators and key factors may influence these decisions. Programmatic implications may differ between cases of PrEP cycling driven by changes in HIV risk and cycling due to contextual barriers despite sustained high risk of HIV acquisition. While the former requires passive follow-up, more intensive tracing and counselling support may be needed for those at sustained risk engaging in cycling.

Through qualitative work, our study sought to understand PrEP knowledge and acceptability among FSW in eThekwini, South Africa, as well as motivators and challenges to PrEP uptake and long-term use. The aim of this analysis was to characterize key social ecological factors contributing toward PrEP cycling among FSW in the context of high HIV risk.

## Methods

### Study setting and data collection

Data were collected in eThekwini, South Africa as a supplement to the *Siyaphambili* randomized trial aiming to compare the effectiveness and feasibility of two adaptive HIV treatment support strategies, individualized case management and decentralized treatment provision, among FSW living with HIV who were virally unsuppressed at baseline [21]. This supplement aimed to adapt individualized case management for HIV-negative FSW initiating PrEP. Data presented here are from the formative qualitative work implemented as the first phase of the supplement. The *Siyaphambili* study and this supplement were implemented through TB HIV Care, a South African non-profit organization providing HIV prevention and treatment services to FSW. Formative data collection began in January 2020 and was delayed due to COVID-19-associated restrictions in March 2020; activities resumed in July, 2020 and were completed in October, 2020. Semi-structured in-depth interviews (IDIs) and key informant interviews (KIIs) were conducted with 36 FSW and 12 key informants (KIs). Criterion-based sampling was used to ensure a variety of perspectives and PrEP experience levels from FSW. Participants were recruited to fulfil three key groups: 1) FSW who had recently tested negative for HIV and were PrEP eligible but had never used PrEP (n = 12); 2) FSW who had initiated PrEP but did not return for their one-month refill (n = 8); and 3) FSW who had initiated PrEP

and had received their one-month refill (n = 16). PrEP mobile outreach staff from TB HIV Care identified and referred FSW engaged in TB HIV Care HIV prevention services for recruitment based on clinical records documenting patterns of PrEP use. Most referrals were made in real-time immediately following clinical encounters, with interviewers available on-site to screen and enrol participants. FSW were recruited and IDIs conducted at both sex work venues where decentralized HIV prevention services were being delivered and at TB HIV Care's drop-in centre. Prior to enrolment, qualitative interviewers screened all participants to confirm eligibility. KIs included nurses, counsellors, and FSW peer educators employed by TB HIV Care, along with sex work venue managers.

Semi-structured interview guides explored themes of perceived HIV risk, PrEP knowledge and perceptions, barriers and facilitators to PrEP uptake and adherence, experiences with TB HIV Care services, and input on a future peer case management intervention. Separate interview guides were used for FSW and key informants in order to tailor specific questions for HIV prevention service user and PrEP service implementer perspectives. In interviews conducted between July and October 2020, participants were also asked about COVID-19 impacts on their daily lives and on their PrEP use when applicable. Interviews were audio-recorded and conducted in English or isiZulu. English interviews were directly transcribed while isiZulu interviews were simultaneously translated into English and transcribed. Following translation and transcription, the written English translation was compared to the isiZulu audio to ensure accuracy. All interviews were conducted in-person by co-author Mkhize and ranged from approximately 20 to 60 minutes in length. FSW were given 100 ZAR (~$7 USD) as compensation for their time and any transport costs associated with participation in IDIs.

## Data analysis

Interviewers filled out rapid analysis forms (RAFs) within 24 hours of each interview to capture initial impressions and summarize each participant's responses. RAFs were reviewed by co-authors including two master's students, the research program coordinator, and the principal investigator. Debriefing sessions were held regularly with the interviewer (co-author Mkhize) to discuss emerging themes and provide feedback to ensure data collection and analysis were iterative. A detailed analysis was conducted using a codebook that was iteratively developed based on key themes informed by the interview guides, supplemented with inductive codes drawn from preliminary readings of transcripts. All transcripts were double coded by two co-authors (LS, SR) in ATLAS.ti 8 [22] and coding discrepancies were rectified through discussions between coders with additional input from two additional co-authors (CAC, SRS). Code reports were generated and reviewed and code-specific sub-themes were summarized. These sub-themes were then organized into overarching key themes identified across all data and codes. Finally, these overarching key themes were organized into social ecological levels guided by Baral et al.'s modified social ecological model for HIV risk [23] during the post-coding analysis phase.

## Results

Demographic information is summarized for the 36 FSW who participated in IDIs in Table 1. FSW ranged from ages 19–56 years, with 83% of participants identifying as single (n = 30/36). Women worked in a variety of venue types, including brothels (n = 11/36), hotels/guest houses (n = 10/36), or private homes (n = 7/36). FSW reported working during both day and night-time hours (n = 25/36), with slightly more than a quarter of women working only during daytime hours (n = 10/36). KIs included counsellors and nurses who work with FSW (n = 4/12),

**Table 1. Demographics of female sex workers (FSW) qualitative participants across pre-exposure prophylaxis (PrEP) experience levels.**

| | FSW who never used PrEP (n = 12) | FSW who initiated PrEP but did not return for 1-month refill (n = 8) | FSW who initiated and did return for 1-month refill (n = 16) | Total (n = 36) |
|---|---|---|---|---|
| Demographics | n (%) | n (%) | n (%) | n (%) |
| **Age (years)** | | | | |
| 18–25 | 2 (16.7) | 5 (62.5) | 5 (31.3) | 12 (33.3) |
| 26–30 | 5 (41.7) | 1 (12.5) | 2 (12.5) | 8 (22.2) |
| 31–35 | 2 (16.7) | 0 (0) | 2 (12.5) | 4 (11.1) |
| 36–40 | 2 (16.7) | 0 (0) | 5 (31.3) | 7 (19.4) |
| 41–45 | 0 (0) | 0 (0) | 1 (6.3) | 1 (2.8) |
| 45+ | 1 (8.3) | 2 (25.0) | 1 (6.3) | 4 (11.1) |
| **Relationship status** | | | | |
| Single | 10 (83.3) | 7 (87.5) | 13 (81.3) | 30 (83.3) |
| Married | 2 (16.7) | 1 (12.5) | 1 (6.3) | 4 (11.1) |
| Separated/ divorced | 0 (0) | 0 (0) | 2 (12.5) | 2 (5.6) |
| **Venue type** | | | | |
| Private home | 4 (33.3) | 2 (25.0) | 1 (6.3) | 7 (19.4) |
| Brothel | 1 (8.3) | 1 (12.5) | 9 (56.3) | 11 (30.6) |
| Bar or club | 0 (0) | 1 (12.5) | 0 (0) | 1 (2.8) |
| Street/park/ Garden | 0 (0) | 2 (25.0) | 2 (12.5) | 4 (11.1) |
| Hotel/guest house | 5 (41.7) | 2 (25.0) | 3 (18.8) | 10 (27.8) |
| Massage parlor | 1 (8.3) | 0 (0) | 1 (6.3) | 2 (5.6) |
| Lodge | 1 (8.3) | 0 (0) | 0 (0) | 1 (2.8) |
| **Primary work hours** | | | | |
| Daytime | 6 (50) | 3 (37.5) | 1 (6.3) | 10 (27.8) |
| Nighttime | 1 (8.3) | 0 (0) | 0 (0) | 1 (2.8) |
| Both | 5 (41.7) | 5 (62.5) | 15 (93.8) | 25 (69.4) |

FSW peer educators (n = 3/12), venue/sex work managers (n = 4/12), and a social auxiliary worker working with youth and adolescents (n = 1/12).

PrEP cycling was common across participants, but specific experiences varied between participant groups. Of the eight FSW participants who had not returned for their one-month PrEP refill, one woman had recently reinitiated PrEP after temporarily discontinuing, while the remaining seven were not on PrEP at the time of interview. However, six of the seven women reported that they intended to reinitiate PrEP in the near future. Of the 16 FSW interviewed who had returned for their one-month PrEP refill, all were still on PrEP at the time of interview. Within this group, 11 FSW had never cycled off PrEP, while the other five indicated that they had temporarily cycled off PrEP in the past and later reinitiated.

Key themes that emerged at the individual level included temporary discontinuation of PrEP driven by negative experiences with side effects, incomplete knowledge of PrEP side effects, and limited understanding of how to use PrEP correctly (Table 2). Among FSW participants who were eligible for PrEP but had never tried it, knowledge of PrEP was more limited. Many of these women knew PrEP prevented HIV and was taken as a daily pill, and some were aware of side effects from friends who took PrEP. However, none of these women reported knowledge of the ability to stop and start PrEP. Among FSW who either had already reinitiated

**Table 2. Quotes illustrating perspectives from female sex workers (FSW) and key informants (KIs) at multiple social ecological levels.**

| | |
|---|---|
| **Individual-Level** | *"Some understand but there are some who-, you find a person-, because there are some who would take it and then stop it. But when you follow-up and continue talking to them then they restart again."* **FSW peer educator**<br>*"I think you have to be deep when you explain that it's optional because when they stop you find that it's painful when one comes back and find that she's already positive. I think health education about the seasonal option should be deep. So that you stop it when you really see that you are not at risk."* **Nurse working with FSW** |
| **Social & sexual networks** | *"Our clients don't stay. They move and there's a time here she would visit home and when she visits home she would stop. When they come back to business they start again. . .One would be afraid to explain to the family, then one would leave behind the pills and say 'I have left the pills I was not in business but now I am back' and others would call you and say 'I am back, I was not around I was home'. One would say 'I left the pills, I am not taking them because they will ask me at home why I am taking the pills, then it would be difficult to explain why I am taking these pills.'"* **FSW peer educator** |
| **Organizational/ Community** | *"What led me to take it only for 1 month is because I came here [to the mobile clinic], they gave me then I left. Then I didn't know that if I were to go to the clinic and ask for it whether I'll get it or not because I registered here. That is why, then I thought I will wait for you to come back then I'll take it again."* **FSW who initiated PrEP & discontinued** |
| **Structural/Public Policy** | *"This COVID thing, let me say especially the lockdown. It has a huge impact in people not being reachable. And that those places where they work, they were shut down. So when they closed down they left. . .Some would call. . .and say 'I am back, can I please have my pills' but because it's been long we have to start from scratch and test them again so that we can be sure that she is negative before she continues."* **FSW peer educator** |

or planned to reinitiate PrEP, high perceived HIV risk was a key motivator to restart PrEP. At the social and sexual network level, stigma tied to both sex work and PrEP use and high mobility emerged as common challenges, while a desire to live a long life and provide for their children and concerns related to condoms breaking or condomless sex were key motivators to resume PrEP use. At the organizational level, challenges related to service delivery and accessibility were common reasons for temporary discontinuation, while at the structural level, COVID-19 restrictions and criminalization of sex work were barriers to consistent PrEP use. Fig 1 shows results organized into the modified social ecological framework for HIV risk [23].

## Individual-level factors impacting PrEP cycling

**HIV risk perception.** The majority of FSW recognized that they were at high risk for HIV infection, although some reported feeling that their risk was reduced or even low due to consistent condom use and/or PrEP use. Despite self-reported high perceived HIV risk, FSW who temporarily discontinued PrEP did not cite changes in HIV risk as a motivator for stopping PrEP. However, some women did describe their fear of HIV acquisition and the riskiness of sex work as important motivators that led them to resume PrEP use and overcome barriers that had contributed to their initial discontinuation. One woman described how seeing other FSW become infected by HIV motivated her to start taking PrEP more consistently:

*"I didn't take it long. I won't lie. . .I was on and off with it. But now seeing all the girls on the road getting sick and getting infected, I don't want to be that. So I would rather continue to come here every month, collect my medication, make sure that they finish and come back, you know. To keep myself safe because I've got a long way to go, I don't want to die."*

–FSW on PrEP

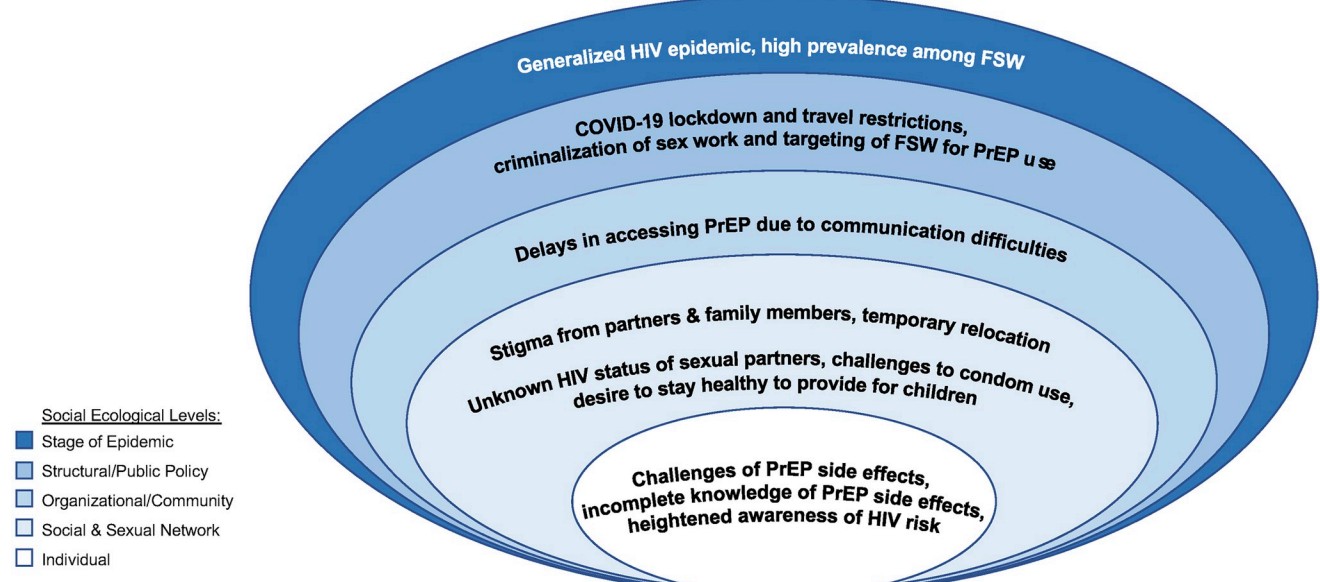

**Fig 1. Modified social ecological model illustrating multi-level factors contributing toward cyclical use of PrEP among FSW as reported by FSW and key informants, including reasons for discontinuing PrEP and motivators to restart PrEP.** Levels shown here are situated in the context of South Africa's generalized HIV epidemic and the particularly high prevalence of HIV among FSW and their clients.

For this participant, seeing first-hand the impacts of HIV on her peers and their health contributed to her own increased HIV risk perception and perceived need for PrEP.

The role of HIV risk in PrEP use decision-making among FSW contrasted with the perspectives shared by KI service providers. KIs reported emphasizing to FSW during counselling sessions the importance of relying on HIV risk as the primary consideration when temporarily stopping PrEP use, as illustrated in this quote:

> *"We tell them 'No, feel free, try PrEP because you can take PrEP and if you feel that you are no longer at risk or you want to go home for 2 months then you can stop it and go back to it when you feel ready'. . .So actually with PrEP you have options, you are free. You have an option to stop taking it and going back."*

–Counsellor working with FSW

Examples given by KIs of appropriate times to stop using PrEP were when FSW were abstaining from sex or temporarily stopping sex work for multiple months or years, which often coincided with trips to their hometown. However, many acknowledged that FSW often stopped for a number of other non-risk related reasons including an incorrect understanding PrEP's preventative effects following discontinuation. One KI described the difficult situation that could arise when FSW stopped taking PrEP while still at high risk for HIV, an experience that was common among KI service providers:

> *"Taking PrEP doesn't mean that you don't use a condom, some would be taking PrEP but when they take a break they would think that it's still in their blood and it's still protecting them. She would continue and engage in unsafe sex, when she can see that there's more problems now she would come back and say 'my sister I want to continue with PrEP' when you test her you find that she is already positive."*

–Nurse working with FSW

These experiences further motivated KIs to emphasize to FSW the importance of HIV risk when making decisions around cycling off PrEP, and to ensure counselling was in-depth when it came to "seasonal" use of PrEP (Table 2).

**Experiences with side effects.** Many FSW who had previously used or were currently on PrEP reported experiencing side effects such as nausea, loss of appetite, and headaches when first initiating PrEP. Among FSW who temporarily discontinued PrEP use, women reported feeling overwhelmed and unable to manage the side effects, leading to PrEP discontinuation within the first week. Some women reported a lack of knowledge of side effects before starting PrEP, either due to incomplete counselling from clinic staff or because they had difficulty remembering what they had been told during the initiation process. This led to a sense of fear for some women as they did not expect to experience side effects and were concerned PrEP was negatively impacting their health. One FSW described her experience stopping PrEP due to side effects in the quote below:

> *"PrEP, I started taking it now because in the beginning I did take it and it made me sick, I was vomiting and lost appetite. . .Then I stopped a bit, then these ladies [from the mobile clinic] came yesterday and I told them my story about what happened. Then they said maybe there was a problem with me that led to me vomiting. Then they said I should try again to take them and see if there is any difference."*

–FSW who initiated PrEP & discontinued

Many FSW felt confident in their ability manage side effects and safe to return to PrEP after talking to clinic staff again and receiving additional counselling on side effects and their one to two week timeframe along with medication to address them when needed.

## Social and sexual network factors impacting PrEP cycling

**Temporary relocation.** Frequent relocation was common among FSW for multiple reasons. Some described moving between cities in search of more clients and higher demand to maximize their earnings. Many FSW were not originally from eThekwini and would leave the city to go back to their hometown to visit family for months at a time. These instances of temporary relocation posed barriers for long-term PrEP use due to the lack of PrEP availability in many regions, particularly rural areas. While some FSW reported notifying clinic staff of their departure ahead of time and receiving a supply of PrEP to take with them, many left without consulting staff, resulting in temporary discontinuation followed by re-initiation on PrEP once the women returned to eThekwini. While levels of risk were variable during travel depending on individual sexual behaviors, participants who discontinued PrEP during these periods reported doing so due to a lack of planning prior to travel or challenges taking pills in these environments rather than conscious assessment of personal risk in consultation with their providers. Perspectives from KI service providers aligned with FSW experiences.

**Interpersonal stigma from family and partners.** FSW reported experiencing or anticipating stigma from family members and partners due to their PrEP use, leading many to avoid disclosing or sharing details on the topic. For many, this stigma arose in the context of temporary relocation when visiting family or long-distance partners, or with steady partners in eThekwini. FSW were reluctant to bring their PrEP with them when visiting home or take pills in front of family, partners, or clients, either because FSW worried they would assume they were

HIV positive and taking antiretrovirals or because of the common perception that PrEP was only for sex workers, as illustrated in this quote:

*"At home they won't understand because they don't know these pills. So that's why I quit in Zimbabwe because my boyfriend won't understand, because they are the same with the HIV...pills, the bottles are the same and so they won't understand. That's why I quit but I'll start very soon."*

–FSW who initiated PrEP & discontinued

The compounded stigma caused by the link between PrEP use and sex work was a key barrier that led many FSW to temporarily discontinue PrEP use when visiting family and partners, since most chose not to disclose their sex work to these individuals. Several KI service providers similarly described these dynamics and their impacts on consistent PrEP use (Table 2).

**Social and sexual networks as motivators to resume PrEP.** Interpersonal dynamics within both sexual and social networks acted as motivators for some women to resume PrEP use after discontinuing. Within sexual networks, FSW worried about their continued high HIV risk during periods of discontinuation, especially with clients. Fear of condoms failing or anticipated difficulty negotiating condom use with clients was a strong motivator for restarting PrEP. One FSW who had temporarily discontinued PrEP described her fear of being exposed to HIV if a condom were to break:

*"Then time went by and I stopped [PrEP for] a bit, I won't lie I did stop taking them. After I stopped them I thought 'no it's not good stopping them because mistakes do happen that the condom bursts while having sex with someone, what if the condom bursts with the wrong person?' then I went back to it."*

–FSW on PrEP

This quote illustrates the connection between sexual network dynamics and perceived HIV risk, both of which informed multiple women's decisions to return to PrEP.

Social networks, particularly familial relationships, also played an important role in motivating women to return to PrEP. Among both women who had already reinitiated PrEP and those who intended to do so in the near future, several participants reported being motivated to restart PrEP to ensure they would live long, healthy lives and stay alive for their children. Some also noted that they had seen family members or peers die of AIDS, which further motivated them to do everything they could to prevent HIV.

## Organizational and community-level factors

**Access issues.** Both KIs and FSW described challenges related to PrEP access and service delivery. Service providers reported difficulty locating FSW at work sites and contacting FSW given how frequently women changed phone numbers. This led to delays in delivering PrEP refills to some women, resulting in temporary discontinuation until contact was successfully made. FSW similarly described delays in accessing PrEP from their perspective, even when they had staff's contact information:

*"I left and went home...Because I have to take food home for my children, and my date to collect PrEP passed and I couldn't get hold of the nurses. The nurses couldn't get hold of me when they tried to call, I was far [away]."*

–FSW who initiated PrEP & discontinued

Some FSW were unfamiliar with the mobile clinic delivery schedule or did not have mobile staff's phone numbers. FSW also reported running out of PrEP while waiting for mobile clinic staff to return to their work venue to provide refills (Table 2). Others described difficulty getting in contact with service providers even when they did proactively reach out to obtain PrEP refills. In these cases, FSW were able and motivated to resume PrEP use once they eventually connected with service providers and received PrEP. These access and communication issues specifically occurred for FSW relying on venue-based delivery of PrEP given variability in mobile delivery locations and schedules necessitating prior communication. While most FSW were aware of the availability of PrEP at the drop-in centre, some were unsure if they could take advantage of these services given they had originally opted for mobile delivery.

## Structural and policy-level factors impacting PrEP cycling

**Public policy environment and rollout of PrEP.**   KIs reported stigma surrounding HIV, PrEP use, and sex work as a common experience for FSW. Many noted that the initial rollout of PrEP in South Africa which targeted key populations including sex workers contributed to this stigma and the common misconception that PrEP was only for sex workers. The lack of widespread PrEP availability at public clinics further reinforced this assumption that it was only meant for sex workers:

> "Some of the government clinics and some places that are public sector they don't have PrEP, so [FSW] think that it's specifically for the organizations. So that is exposing them to their family members that 'why here only?'. You find that in the family, it stigmatizes them on people who don't know what kind of job they do. It becomes as if PrEP is for people who are high risk, who are sex workers. So it means somehow it exposes them because it's not available in public clinics."

–FSW peer educator

KIs and FSW alike noted that the lack of national media campaigns or awareness of what exactly PrEP was further encouraged incorrect and often stigmatizing assumptions about those who use PrEP. For FSW, these policy-level factors contributed to the interpersonal stigma they experienced from family and partners as described above as well as misinformation spread among peers, leading to PrEP cycling.

**COVID-19 policies.**   Government-mandated COVID-19 lockdowns and travel restrictions had significant impacts on FSWs' ability to both work and access PrEP. The closure of some sex work venues including bars and restaurants during lockdown caused many FSW to relocate to their hometowns, unable to return to eThekwini to obtain PrEP refills once they ran out due to travel restrictions. KI service providers similarly described the challenges government restrictions posed to PrEP adherence and follow-up with clients (Table 2).

Those who remained in eThekwini faced challenges as well, with some reporting being barred by police from leaving their homes to collect PrEP refills:

> "The police just said 'listen here, we don't even want to see you because you are going to spread Corona' we said no we have to get our things so that we can drink [PrEP], they said 'no you can't'. That's how we stopped and then we waited to hear from the nurses when they are available because they said 'even your nurses are not available' then we said we'll hear from the nurses when they are back to help us."

–FSW who initiated PrEP & discontinued

KI service providers also experienced difficulties from their side, particularly in locating women who had moved or whose work sites were shut down. This was particularly challenging for mobile clinic staff, who would normally visit sex work venues to provide HIV services.

## Discussion

These data demonstrate the complex interplay between multi-level factors within the modified socioecological model impacting sustained PrEP use and contributing toward PrEP discontinuation and re-initiation among FSW. Perceived HIV risk, interpersonal relationships and stigma, service delivery and accessibility, and government policies impacted PrEP cycling and discontinuation, emphasizing the importance of understanding contextual factors impacting FSW across all social ecological levels. To date, much of the research on the dynamics of HIV risk and PrEP use among FSW has focused on the relationship between individual-level risk perception and motivation to adhere to PrEP [24–26]. In the context of PrEP cycling, accurately assessing personal risk is crucial to practicing this behaviour safely. A recent analysis of the relationship between PrEP adherence and HIV risk among adolescent girls and young women participating in the HPTN 082 trial indicated that some women receiving an adherence intervention were more likely to cycle their PrEP use according to risk [27]. However, in general and including in this study, risks are often high in cases of low adherence as well [27]. Our findings show that service providers emphasized the importance of only cycling off PrEP during periods of low risk. However, instances of PrEP cycling among FSW in our sample were more often driven by external factors outside of individuals' control, rather than due to perceived lack of HIV risk. These differences in findings may be due to FSWs' persistently high HIV risk and unique contextual barriers compared to young women more broadly.

FSW encountered barriers at the individual, social network, community, and policy levels leading to PrEP cycling. Impacts of HIV and sex work related stigma on PrEP uptake among FSW and young women have been described in the literature [28–31]. FSW in our sample experienced high levels of anticipated stigma with family and partners, often tied to fear of associations between PrEP and sex work, posing significant barriers to not only PrEP uptake but also sustained use. FSW and KIs alike suggested that increased media coverage and informational campaigns on PrEP and its use to prevent HIV would help to alleviate these forms of stigma and misinformation. Additionally, the advent of long-acting injectable (LAI) PrEP may help to address the interpersonal stigma FSW described by making PrEP use more discrete as has been described in qualitative analyses of data from LAI PrEP [32] and ART [33–35] clinical trials. At the service delivery level, challenges arose with communication when coordinating PrEP mobile delivery at sex work venue sites. Difficulty contacting FSW for PrEP drop-offs caused by complexities around phone use and ownership among FSW are consistent with qualitative findings from the *Siyaphambili* trial [36]. Furthermore, access issues leading to PrEP cycling were exacerbated by COVID-19 lockdown measures. Experienced or perceived challenges of health care accessibility and lack of compatibility with work schedules have been characterized among FSW and the population overall [37–40]. Accounts from FSW and KIs indicate that these community-level barriers were further compounded during the pandemic through policy-level factors such as travel restrictions. Police-enforced lockdowns highlight the need for policy approaches that ensure access to essential health care services is not restricted by pandemic response measures.

In addition to the multi-level barriers challenging PrEP use, motivating factors were found at the individual and social and sexual network levels. FSW reports of being motivated to

resume PrEP due to the unknown HIV status of partners, difficulty negotiating condoms, and a desire to maintain their overall health are consistent with findings on motivations for initial uptake of PrEP among FSW [20, 41] and women more broadly [17]. Our findings further add to this literature by characterizing a motivation among some FSW to re-initiate PrEP not only for the sake of their own wellbeing and future, but also to ensure they live long enough to provide for their children. Importantly, most FSW who temporarily discontinued PrEP in our sample were at high risk for HIV throughout periods of cycling, but motivating factors described above including recognition of their persistently high HIV risk ultimately led them to restart PrEP despite challenges. Our results illustrate the impact of barriers and facilitators across multiple socioecological levels on PrEP cycling behaviours.

There are several limitations to this study. Recruitment of FSW was completed during daytime working hours, which may have contributed to the high proportion of FSW in our sample who reported working during the day or both day and night-time hours. Because of this, themes specific to FSW only working night-time hours such as conflicts between recommended timing of pill-taking and work hours may not have fully emerged. Finally, the collection of data taking place partially during the COVID-19 pandemic may have influenced participant perspectives and certain key takeaways in our results given the significant impacts the pandemic had on sex work operations and access to PrEP services.

## Conclusion

PrEP cycling can provide FSW with the power to make informed choices when it comes to their own HIV prevention decisions in the context of changing risk levels. However, our study highlights how FSW are often unable to make these decisions solely based on risk assessments, leaving them vulnerable to HIV acquisition during periods of temporary PrEP discontinuation. Despite many FSW being concerned about their HIV risk and health, multilevel barriers including stigma, fear of disclosing PrEP use, side effects, and access issues led to many women discontinuing PrEP until these barriers were ameliorated or overcome. These instances of temporary discontinuation occurred regardless of actual HIV risk. Interventions addressing these barriers such as social norms campaigns around PrEP use and tailored support strategies to encourage sustained PrEP use can serve to address these barriers and ensure that FSW feel empowered within their environments to make risk-informed decisions to continue with or cycle off PrEP based on their own health needs.

## Acknowledgments

We would like to thank the women and key informants in eThekwini who took the time to participate in these interviews and shared with us their experiences and professional insights. We are also grateful to the TB HIV Care outreach staff, who helped to recruit participants for this activity. The contents expressed here are the sole responsibility of the authors and may not represent the views of the NIH. This research was presented as a poster presentation at the 11th International AIDS Society Conference on HIV Science (IAS 2021), July 18–21, 2021.

## Author Contributions

**Conceptualization:** Lillian M. Shipp, Carly A. Comins, Deliwe Rene Phetlhu, Harry Hausler, Stefan D. Baral, Sheree R. Schwartz.

**Data curation:** Lillian M. Shipp, Sofia Ryan, Carly A. Comins, Ntambue Mulumba, Vijaya-nand Guddera.

**Formal analysis:** Lillian M. Shipp, Sofia Ryan, Carly A. Comins, Sheree R. Schwartz.

**Funding acquisition:** Harry Hausler, Stefan D. Baral, Sheree R. Schwartz.

**Investigation:** Mfezi Mcingana, Deliwe Rene Phetlhu, Harry Hausler, Stefan D. Baral, Sheree R. Schwartz.

**Methodology:** Lillian M. Shipp, Sofia Ryan, Carly A. Comins, Mfezi Mcingana, Deliwe Rene Phetlhu, Harry Hausler, Stefan D. Baral, Sheree R. Schwartz.

**Project administration:** Carly A. Comins, Mfezi Mcingana, Ntambue Mulumba, Vijayanand Guddera, Harry Hausler.

**Resources:** Ntambue Mulumba, Harry Hausler, Stefan D. Baral.

**Supervision:** Ntambue Mulumba, Vijayanand Guddera, Harry Hausler, Stefan D. Baral, Sheree R. Schwartz.

**Validation:** Sofia Ryan, Carly A. Comins, Sheree R. Schwartz.

**Visualization:** Lillian M. Shipp, Carly A. Comins, Sheree R. Schwartz.

**Writing – original draft:** Lillian M. Shipp.

**Writing – review & editing:** Sofia Ryan, Carly A. Comins, Mfezi Mcingana, Ntambue Mulumba, Vijayanand Guddera, Deliwe Rene Phetlhu, Harry Hausler, Stefan D. Baral, Sheree R. Schwartz.

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
