## [Decision Letter · Decision Letter 0]

5 Mar 2024

PONE-D-23-32704PrEP discontinuation, cycling, and risk: Understanding the dynamic nature of PrEP use among female sex workers in South AfricaPLOS ONE

Dear Ms. Shipp,

Thank you for submitting your manuscript to PLOS ONE. After careful consideration, we feel that it has merit but does not fully meet PLOS ONE’s publication criteria as it currently stands. Therefore, we invite you to submit a revised version of the manuscript that addresses the points raised during the review process.

I am pleased to share with you the reviewers' comments. Please make sure that you pay attention to the comments by the reviewers' particularly those that are marked major but also pay close attention to the other non-major comments==============================

We look forward to receiving your revised manuscript.

Kind regards,

Martin Mbonye

Academic Editor

PLOS ONE

Journal Requirements:

Reviewers' comments:

Reviewer's Responses to Questions

**Comments to the Author**

1. Is the manuscript technically sound, and do the data support the conclusions?

Reviewer #1: Yes

Reviewer #2: Yes

2. Has the statistical analysis been performed appropriately and rigorously? 

Reviewer #1: N/A

Reviewer #2: N/A

3. Have the authors made all data underlying the findings in their manuscript fully available?

Reviewer #1: Yes

Reviewer #2: No

4. Is the manuscript presented in an intelligible fashion and written in standard English?

Reviewer #1: Yes

Reviewer #2: Yes

5. Review Comments to the Author

Reviewer #1: See attached.

Reviewer #2: Abstract: the purpose of the study not stated.

Introduction: Well presented

Results: The study presented the results of the primary scientific research

Discussion: I did not see the application of the framework chosen and presented above.

Conclusion: Yes, the conclusion supports the results

Overall: This is a very good manuscript and it address the important aspects that contribution to on HIV transmission.

6. PLOS authors have the option to publish the peer review history of their article (what does this mean?). If published, this will include your full peer review and any attached files.

Reviewer #1: No

Reviewer #2: No

---

## [Author Response · Author response to Decision Letter 0]

19 Apr 2024

Reviewer #1:

This is a very interesting study that provides valuable insights into PrEP cycling among female sex workers in South Africa. The manuscript is well written but there are a few major and minor comments that need to be addressed to further improve it.

Major

1. The introduction is well written and the issue of PrEP cycling despite sustained high risk of HIV acquisition is clearly shown. The aim of the study is also clear although there seems to be a slight disconnect between the gap and the aim. You indicate in line 61-62 that “less is known about whether women who discontinue PrEP choose to later re-initiate and what potential motivators and key factors may influence these decisions”. The aim does not seem to be specific to the PrEP cyclers, but it would be good to have this connection between the gap and the aim by adding a word on ‘PrEP cycling’ to your aim in line 66-68 to keep focus.

We thank the reviewer for this comment. In response, we have added the following clarifying text to line 68 to distinguish between the overall purpose of the qualitative study and the aim of the specific analysis presented in this manuscript: 

“The aim of this analysis was to characterize key social ecological factors contributing toward PrEP cycling among FSW in the context of high HIV risk.”

2. In your methods, a) Give a little bit more detail on the implementation of the Siyaphambili trial and how this supplement study contributed to the siyaphambili trial for the reader to know how this study fitted in or how the two studies were connected (line 72-74).

We appreciate this comment from the reviewer. We have added the following text to lines 74-77 to provide additional context on the design of the Siyaphambili trial and further explanation of how this supplement connected to the larger trial: 

“Data were collected in eThekwini, South Africa as a supplement to the Siyaphambili randomized trial aiming to compare the effectiveness and feasibility of two adaptive HIV treatment support strategies, individualized case management and decentralized treatment provision, among FSW living with HIV who were virally unsuppressed at baseline (21). This supplement aimed to adapt individualized case management for HIV-negative FSW initiating PrEP. Data presented here are from the formative qualitative work implemented as the first phase of the supplement.”

b) Was there further screening/assessment of eligibility when the FSW were referred. If not, how did you minimize sample bias considering that referrals were done by the organization staff, whose services were somehow being assessed through FSW experiences? Add some bit of detail to the recruitment processes, since some participants were those who did not initiate/get one-month refills.

Thank you for this comment. We have added text to line 87-93 to describe how eligibility screening was conducted, first by programme staff and then confirmed by interviewers prior to informed consent. Individuals referred by the programme were seeking HIV prevention services, which includes HIV testing, condoms, and PrEP, allowing for identification of FSW within each category of participants. In the case of those who had never used PrEP or had not returned for their 1-month refill, these individuals were still engaged with the programme and accessing other prevention services. Given that we used a purposive sampling method (criterion-based sampling) the aim was to capture a wide range of perspectives and experiences with PrEP within the FSW population by seeking diversity rather than representativeness. In order to obtain a diverse set of experiences among participants, we also made an effort to sample from a variety of sex work venues that included multiple venue types, including indoor and outdoor settings, as is illustrated in Table 1. Furthermore, the purposive sampling method inherently introduces potential bias during the participant selection process, however screening by interviewers ensured all participants were eligible and met the criteria we were sampling on.

The added text in lines 88-93 is as follows:

“PrEP mobile outreach staff from TB HIV Care identified and referred FSW engaged in TB HIV Care HIV prevention services for recruitment based on clinical records documenting patterns of PrEP use. Most referrals were made in real-time immediately following clinical encounters, with interviewers available on-site to screen and enrol participants. FSW were recruited and IDIs conducted at both sex work venues where decentralized HIV prevention services were being delivered and at TB HIV Care’s drop-in centre. Prior to enrolment, qualitative interviewers screened all participants to confirm eligibility.”

c) At what point did the mobile staff refer them for interviews and where would they find them (did they find and give them referrals during their mobile outreaches, or did they give phone contacts or other locator information to the study staff? Please indicate these details (line 83-84

Thank you for this comment. Mobile staff referred potential participants for interviews once clinical interactions (such as HIV testing, PrEP dispensing, STI testing) were completed. Most often, interviewers were at the outreach sites or drop-in centre and able to conduct interviews immediately upon referral. This approach helped to minimize the risk of closing potential participants to follow-up if contact could not be made following referrals. It also ensured a diversity of venues were represented in the sample. In some cases where immediate screening and enrolment was not possible, participant contact information was given to interviewers by the outreach staff and interviews were conducted within a few days of referral. Text added to lines 88-93 (quoted above in response to comment 2b) elaborates on these details.

d) Line 84-85 indicates that interviews were conducted at sex work venues or HIV TB Care’s drop-in centre. What guided you in deciding the interview venue for the different interviews.

Thank you for this question. In most cases, interviews were conducted at the site of recruitment, immediately following referral in order to make the process more efficient and convenient for participants. These venues were locations where the outreach teams deliver decentralized HIV prevention services. In cases where interviews were done at a later date following recruitment, interviewers coordinated with potential participants to decide on a convenient location for the interviews at either an outreach site or the drop-in centre.

e) Did you use the same interview guide for FSW and the Key informants? Please indicate.

Thank you for this question. We used separate interview guides for these two groups in order to tailor interview questions and probes to the FSW and program implementer perspectives. We have added the following text to lines 97-99 to indicate this:

“Separate interview guides were used for FSW and key informants in order to tailor specific questions for HIV prevention service user and PrEP service implementer perspectives.”

f) Line 95 indicates giving incentives, which do not conform with the GCP guidelines. I would suggest that we call this a compensation (it could be for their time/transport)

Thank you for flagging this important point. We have updated the language as suggested to more accurately describe the intention of these payments as reimbursements for time spent participating in the interviews. The updated text is as follows:

“FSW were given 100 ZAR (~$7 USD) as compensation for their time and any transport costs associated with participation in IDIs.”

g) Line 92-93 is a bit confusing, when you say isiZulu interviews were simultaneously translated to English, I get a feeling there was no isiZulu transcript but a direct audio translation from the local language to English. So, this comparison between the original isiZulu and English transcripts is a bit confusing. Please cross-check this. Also indicate who conducted the interviews.

Thank you for this comment. We agree with the reviewer that the description of the transcription and translation process included in the text may cause confusion. In this case, the transcriber did do a direct audio translation from isiZulu into English and then transcribed in English, and no isiZulu transcript was created prior to translation. Once English translations were transcribed, the transcriber then listened back to the isiZulu audio and cross-compared with English transcripts to ensure accuracy. We have clarified this in the text in lines 106-109:

“English interviews were directly transcribed while isiZulu interviews were simultaneously translated into English and transcribed. Following translation and transcription, the written English translation was compared to the isiZulu audio to ensure accuracy.”

h) You indicate in the analysis (line 99-100) that the larger team reviewed RAFs. Who constituted the larger team? Please show a brief composition of the study team. Also remember to indicate a statement on authors’ contributions.

Thank you for this comment. In this case, the study team members reviewing RAFs consisted of co-authors Shipp, Ryan, Comins, and Schwartz. Roles were as follows: master’s student research assistants (Shipp, Ryan); research program coordinator (Comins); and principal investigator (Schwartz). Debriefing sessions consisted of the above co-authors and co-author Mkhize, who conducted the interviews. A statement on author contributions has been included via the submission portal. We have added text to the methods section as follows:

“RAFs were reviewed by co-authors including two master’s students, the research program coordinator, and the principal investigator. Debriefing sessions were held regularly with the interviewer (co-author Mkhize) to discuss emerging themes and provide feedback to ensure data collection and analysis were iterative.”

i) In your data analysis, you mention RAFs, I presume these followed a specific thematic format since you indicated that your analysis was done deductively and inductively, and that your results followed/ were guided by Baral’s modified social ecological model (MSEM). In that respect, you need to clearly indicate at what point in your research you employed the model. Were your data tools designed in respect to the model? 

Thank you for this comment. In-depth interview guides and RAFs were not designed with this specific model. However, attention was given to include questions and topics targeting multiple socioecological levels. The MSEM was applied during post-coding analysis to structure and contextualize key themes that emerged from the data during the coding process. We have added text to line 129 to clarify the stage at which the MSEM was applied.

j) Line 101 on data analysis, I suggest that you indicate this as ‘detailed analysis’ since you had already mentioned analysing your data rapidly, so that it does not get confusing. Line 105-109 lacks clarity of how the analysis process was done. The reviewing of coded transcripts and the summarizing of sub-themes is quite confusing. Indicate the overarching themes that you identified. 

Thank you for this comment. We have updated the language to the data analysis description to distinguish between rapid analysis forms which were used to inform iterative data collection, and detailed analysis which was performed using coding and post-coding analysis techniques. Sub themes, including barriers and facilitators to PrEP use and factors contributing to PrEP cycling, emerged across multiple levels (individual, interpersonal, societal, and structural). These emergent themes were then grouped into overarching key themes and organized into the Baral et al. MSEM to reflect the multilevel nature of these factors. The overarching key themes are those reflected in Figure 1 as organized into the MSEM. The updated text is as follows:

“Code reports were generated and reviewed and code-specific sub-themes were summarized. These sub-themes were then organized into overarching key themes identified across all data and codes. Finally, these overarching key themes were organized into social ecological levels guided by Baral et al.’s modified social ecological model for HIV risk (23) during the post-coding analysis phase.”

3. Results- I would suggest that you systematically structure your results following the MSEM like you indicated in your introduction and analysis. Starting with the text on key issues that emerged (line 132- it should actually start from line 129 as knowledge is an individual level factor), I suggest that you clearly organize these results under the different levels of the MSEM, where they each fit, followed with supporting quotes. In that case, you will not need a separate table for quotes, you could instead provide a table for KI demographics. 

Thank you for this suggestion. We have restructured the paragraph describing key themes that emerged to more clearly follow the levels of the MSEM in order to ensure consistent application of the framework. While we understand the value in including quotes integrated directly into the text and have taken this approach within the subsections of the results, we feel that organizing some key quotes into a table helps to make the results section more concise while still ensuring relevant data are presented. We have moved this table up to the key themes paragraph from its original placement in the individual-level subsection given its relevance to multiple levels of the MSEM. KI roles are reported in the results in order to illustrate the variety of professional perspectives included in the sample. Additional demographics are not reported in order to protect the anonymity of KI participants given the small number of staff employed within each role at the TB HIV Care eThekwini site.

The edited text of the paragraph describing key themes is as follows:

“Key themes that emerged at the individual level included temporary discontinuation of PrEP driven by negative experiences with side effects, incomplete knowledge of PrEP side effects, and limited understanding of how to use PrEP correctly (Table 2). Among FSW participants who were eligible for PrEP but had never tried it, knowledge of PrEP was more limited. Many of these women knew PrEP prevented HIV and was taken as a daily pill, and some were aware of side effects from friends who took PrEP. However, none of these women reported knowledge of the ability to stop and start PrEP. Among FSW who either had already reinitiated or planned to reinitiate PrEP, high perceived HIV risk was a key motivator to restart PrEP. At the social and sexual network level, stigma tied to both sex work and PrEP use and high mobility emerged as common challenges, while a desire to live a long life and provide for their children and concerns related to condoms breaking or condomless sex were key motivators to resume PrEP use. At the organizational level, challenges related to service delivery and accessibility were common reasons for temporary discontinuation, while at the structural level, COVID-19 restrictions and criminalization of sex work were barriers to consistent PrEP use. Figure 1 shows results organized into the modified social ecological framework for HIV risk (23).”

Minor 

1. The predetermined criterion in the methods section line is well indicated to show the categories of FSWs that were included. You however need to show how the PrEP outreach staff identified/arrived at the participants. For example, did they scan through the different records, etc. 

Thank you for this comment. We have clarified the recruitment process in the methods section as described in response to major comment 2b.

2. Line 229-239 in your results, you indicate stigma as a key motivator to temporarily discontinue PrEP. I would think that anything that pushes one to discontinue PrEP, would be a demotivator, barrier or challenge.

Thank you for this comment. We agree with the reviewer and have changed the wording in this line to clarify this as follows:

“The compounded stigma caused by the link between PrEP use and sex work was a key barrier that led many FSW to temporarily discontinue PrEP use when visiting family and partners, since most chose not to disclos

---

## [Decision Letter · Decision Letter 1]

2 Sep 2024

PrEP discontinuation, cycling, and risk: Understanding the dynamic nature of PrEP use among female sex workers in South Africa

PONE-D-23-32704R1

Dear Dr. Shipp,

We’re pleased to inform you that your manuscript has been judged scientifically suitable for publication and will be formally accepted for publication once it meets all outstanding technical requirements.

Kind regards,

Martin Mbonye

Academic Editor

PLOS ONE

Additional Editor Comments (optional):

Reviewers' comments:

Reviewer's Responses to Questions

**Comments to the Author**

1. If the authors have adequately addressed your comments raised in a previous round of review and you feel that this manuscript is now acceptable for publication, you may indicate that here to bypass the “Comments to the Author” section, enter your conflict of interest statement in the “Confidential to Editor” section, and submit your "Accept" recommendation.

Reviewer #1: All comments have been addressed

Reviewer #3: All comments have been addressed

2. Is the manuscript technically sound, and do the data support the conclusions?

Reviewer #1: Yes

Reviewer #3: Yes

3. Has the statistical analysis been performed appropriately and rigorously? 

Reviewer #1: N/A

Reviewer #3: N/A

4. Have the authors made all data underlying the findings in their manuscript fully available?

Reviewer #1: No

Reviewer #3: Yes

5. Is the manuscript presented in an intelligible fashion and written in standard English?

Reviewer #1: Yes

Reviewer #3: Yes

6. Review Comments to the Author

Reviewer #1: All initial comments to the author(s) have been addressed to my satisfaction. Just one minor observation in the response to comment (h line 112), I do not think it is necessary to mention the name of the interviewer here, initials are enough. Otherwise, thanks for addressing the comments.

Reviewer #3: This was such an interesting qualitative study and was followed by a well written paper. The detailed comments by the reviewers have been addressed and resulted in a thoughtfully articulated paper. The manuscript is technically sound, but to show that the author has adhered to guidelines for reporting qualitative research i would suggest completing the Coreq checklist and either append it or make a statement that it has been adhered to. The checklist can be found at https://cdn.elsevier.com/promis_misc/ISSM_COREQ_Checklist.pdf. I like the fact that demographic data has been presented in a table. while this is not a prerequisite in qualitative research, it is lacking in many qualitative papers where such information is useful in giving the reader a full picture of the participants.

The only other comment I have is regarding the findings. The finding that 'none of these women reported knowledge of the ability to stop and start PrEP need to be rephrased as it could be misread. Rephrasing it to mean that "None of the participants demonstrated awareness of the option to start or stop PrEP." or similar wording would be better. This finding shows that participants lacked knowledge about the flexibility of managing their PrEP regimen. Also, when reading through the findings, this seems to me to be THE KEY finding as it has consequences how women navigate stopping and starting and the stress they go through when this happens, thinking that they cause harm to themselves or that they are careless and will be perceived as such when they return to the clinic. Without this knowledge/awareness, women may feel constrained or lack control over their own preventive care. Women who are aware of this flexibility, thus have a sense of agency.

7. PLOS authors have the option to publish the peer review history of their article (what does this mean?). If published, this will include your full peer review and any attached files.

Reviewer #1: No

Reviewer #3: No

---

## [Editor Report · Acceptance letter]

16 Sep 2024

PONE-D-23-32704R1 

PLOS ONE

Dear Dr. Shipp, 

I'm pleased to inform you that your manuscript has been deemed suitable for publication in PLOS ONE. Congratulations! Your manuscript is now being handed over to our production team.

Kind regards, 

on behalf of

Dr. Martin Mbonye 

Academic Editor

PLOS ONE